# Towards the Development of Long Circulating Phosphatidylserine (PS)- and Phosphatidylglycerol (PG)-Enriched Anti-Inflammatory Liposomes: Is PEGylation Effective?

**DOI:** 10.3390/pharmaceutics13020282

**Published:** 2021-02-19

**Authors:** Miriam E. Klein, Max Rieckmann, Daniel Sedding, Gerd Hause, Annette Meister, Karsten Mäder, Henrike Lucas

**Affiliations:** 1Faculty of Biosciences, Institute of Pharmacy, Martin Luther University Halle-Wittenberg, 06120 Halle (Saale), Germany; miriam.klein@pharmazie.uni-halle.de (M.E.K.); karsten.maeder@pharmazie.uni-halle.de (K.M.); 2Mid-German Heart Center, Department of Cardiology, University Hospital, Martin Luther University Halle-Wittenberg, 06120 Halle (Saale), Germany; max.rieckmann@uk-halle.de (M.R.); daniel.sedding@uk-halle.de (D.S.); 3Biocenter, Martin Luther University Halle-Wittenberg, 06120 Halle (Saale), Germany; gerd.hause@biozentrum.uni-halle.de; 4Faculty of Biosciences, IWE ZIK HALOmem and Institute for Biochemistry and Biotechnology, Martin Luther University Halle-Wittenberg, 06120 Halle (Saale), Germany; annette.meister@chemie.uni-halle.de

**Keywords:** liposomes, phosphatidylserine, phosphatidylglycerol, nano, PEGylation, pharmacokinetics, biodistribution, fluorescence imaging, optical imaging

## Abstract

The anionic phospholipids (PLs) phosphatidylserine (PS) and phosphatidylglycerol (PG) are endogenous phospholipids with anti-inflammatory and immunomodulatory activity. A potential clinical use requires well-defined systems and for several applications, a long circulation time is desirable. Therefore, we aimed the development of long circulating liposomes with intrinsic anti-inflammatory activity. Hence, PS- and PG-enriched liposomes were produced, whilst phosphatidylcholine (PC) liposomes served as control. Liposomes were either formulated as conventional or PEGylated formulations. They had diameters below 150 nm, narrow size distributions and composition-dependent surface charges. Pharmacokinetics were assessed non-invasively via in vivo fluorescence imaging (FI) and ex vivo in excised organs over 2 days. PC liposomes, conventionally formulated, were rapidly cleared from the circulation, while PEGylation resulted in prolongation of liposome circulation robustly distributing among most organs. In contrast, PS and PG liposomes, both as conventional or PEGylated formulations, were rapidly cleared. Non-PEGylated PS and PG liposomes did accumulate almost exclusively in the liver. In contrast, PEGylated PS and PG liposomes were observed mainly in liver and spleen. In summary, PEGylation of PS and PG liposomes was not effective to prolong the circulation time but caused a higher uptake in the spleen.

## 1. Introduction

In pharmaceutical applications, there is a need for controlled drug delivery systems that act at the target side, over a controlled period of time with the desired amount of drug to be delivered to the target [1]. For parenteral administration, nano-scaled drug delivery systems (nano-DDSs) have gained interest and resulted in several marketed products [2,3]. By delivering their drugs via passive or active targeting [4], they offer valuable therapeutic options, e.g., in the treatment of cancer or inflammation [5].

Phospholipids are important excipients for parenteral applications with an excellent safety profile. They are the main components of liposomes. Several formulation options exist that control biodistribution and drug release. Examples include the use of saturated or unsaturated phospholipids and PEGylation of liposomes, which shields the liposome surface due to conjugation with long polyethylene glycol polymer chains. This shielding leads to an increase in their circulation duration, which can be seen as passive targeting, as was observed for Doxil™/Caelyx™, the first liposomal product that was approved by health authorities [6]. The main component of lecithin is phosphatidylcholine (PC). In addition to PC, natural lecithin contains other phospholipids such as phosphatidylethanolamine (PE), phosphatidylinositol (PI), phosphatidylserine (PS), phosphatidylglycerol (PG) and lysolecithin (LPC) [7]. Some of these minor components are bioactive, for example, the negatively charged molecules PS and PG. Both PS and PG are also endogenous components of organisms, including humans, and play important roles in physiological and pathophysiological processes.

PS is localized at the inner leaflet of plasma membranes [8]. During apoptosis, PS is translocated to the membranes surface, where it serves as an “eat me” signal for phagocytes [9,10,11,12]. This phagocytosis is paralleled by a macrophage-phenotype shift towards anti-inflammatory activity [13]. Pre-clinically, PS-related impacts on ailments such as Alzheimer’s disease [14], ischemia reperfusion [15], osteoporosis [16,17], arthritis [18,19], post myocardial infarction repair [20,21], rhinitis [22] and chronic wound healing [23], were described.

PG is an important component of the lung surfactant [24]. It also presents impacts on the resolution of inflammation, e.g., in infections of respiratory syncytial virus [25], mycoplasma pneumoniae [26] and influenza A virus [27], as well as positive effects on corneal epithelial wound and skin healing [28,29]. Additionally, it might serve as an anti-inflammatory agent, preventing allergic reactions against irrelevant environmental antigens in the lower respiratory tract [30]. Very recently, PG has been suggested as a treatment option against COVID-19-induced inflammation [31].

Existing products with PS and PG include Mepact™ and Ambisome™. It is reported, that both PS and PG increase the uptake by macrophages [32,33,34,35,36]. Although both PLs are currently used as excipients in marketed products, questions concerning their putative anti-inflammatory properties remain open. Especially the impact of important physicochemical parameters such as size, size distribution, surface charge and PEGylation are still unclear. For example, Mepact™ is formulated as multilamellar vesicles (MLVs) [3], whereas Ambisome™ is composed mainly of small unilamellar vesicles (SUVs) [3]. Size-dependent differences in their biodistribution can be expected. It is well known that, depending on their final size, liposomes originate different pharmacokinetics and clearance rates in vivo [37]. Large liposomes, presenting sizes of more than 220 nm, are rapidly cleared via liver and spleen, whereas smaller liposomes reveal longer circulation duration [37]. In the context of intravenous (i.v.) injection, also submicron dimensions are required to avoid capillary blockage or embolism [38]. Furthermore, surface charge and PEGylation are crucial physicochemical characteristics of liposomes. Modification of the surface charge, e.g., the addition of negatively charged PLs, such as PS or PG, increases liposome uptake by macrophages and cells of the mononuclear phagocyte system (MPS) [32]. In contrast, hydrophilic surface modification of liposomes by PEGylation prolongs their circulation duration [6,39].

Despite the fact that interesting and promising data about the biological effects of PS- and PG-based DDS have been reported, the potential clinical translations are difficult because of very broad distribution of particle sizes and the lack of stability data. Therefore, we recently developed PS- and PG-enriched nanodispersions, well defined in terms of physicochemical parameters and investigated their immunomodulatory properties in a mouse peritoneal macrophages assay in vitro [40]. The developed systems presented a dose-dependent reduction of TNFα and were stable over several weeks [40]. With respect to a further translation into clinical use, the next step is the investigation of the in vivo performance in healthy animals.

It was therefore the aim of the present study to compare the biodistribution of the well-defined PS- and PG-enriched nanodispersions with PC and to study the impact of PEGylation in a preclinical mouse model. For the in vivo and ex vivo characterization of liposome pharmacokinetics, fluorescence imaging (FI) was applied. It enables tracking of fluorescent dye loaded nanoformulations in mice non-invasively with high sensitivity and temporal resolution [41]. In particular, we intended to answer the following questions:Are the biodistribution profiles of PS- and PG-enriched nanodispersions in healthy mice comparable?Does PEGylation of PS- and PG-enriched nanodispersions effect their biodistribution in healthy mice?

The most important liposome specifications, including nanometer size, narrow size distribution and defined surface charges, were considered and investigated. Furthermore, to get a comprehensive image of liposome biodistribution in healthy mice, we compared in vivo and ex vivo FI data both qualitatively and quantitatively. Based on the presented data, we plan to perform further pharmacokinetic studies investigating the biodistribution in a murine model of acute myocardial infarction. There, we will address the anti-inflammatory effects of our liposomes in detail and also the clinical benefit of the treatment, i.e., the improvement in heart function.

## 2. Materials and Methods

### 2.1. Materials

The synthetic 1,2-dioleoyl-*sn*-glycero-3-phospho-l-serine sodium salt (DOPS), 1,2-dioleoyl-*sn*-glycero-3-phospho-1′-rac-glycerol sodium salt (DOPG), S100 (a purified natural phosphatidylcholine (PC) derived from soybean) and the PEGylated PL 1,2-distearoyl-*sn*-glycero-3-phosphoethanolamine-*N*-[methoxy(polyethyleneglycol)-2000] sodium salt (MPEG-2000-DSPE) were kindly provided by Lipoid GmbH (Ludwigshafen, Germany). Chemical structures of the PLs are presented in Appendix A. The fluorescent dye, 1,1′-Dioctadecyl-3,3,3′,3′-Tetramethylindotricarbocyanine Iodide (DiR), was purchased from Thermo Fisher Scientific Inc. (Waltham, MA, USA). All other substances are named in the corresponding sections.

### 2.2. Methods

#### 2.2.1. Liposome Preparation

Liposomes were prepared as previously described [40,42]. Briefly, stock solutions of PLs were prepared in chloroform:methanol (4:1, *v*/*v*). Fluorescent dye, DiR, which was dissolved in methanol, was added in a concentration of 10 µg DiR per 5 mg PL to the PL mixture. Organic solvents were evaporated, and the PL films were stored overnight under vacuum. Afterwards, the PL films were hydrated with sterile saline (NaCl, 0.9% (*w*/*v*); pH 6.5; osmolality 308 mOsmol). The large, multi-lamellar liposomes were extruded through 100 nm pore size polycarbonate membranes above the phase transition temperature of the PLs and were diluted to 5 mg PL per mL with sterile saline. Liposomes were stored in amber glass vials at 8 °C protected from light. Before i.v. injection, liposomes were filtrated through 0.2 µm filter membranes to ensure sterility.

#### 2.2.2. Dynamic Light Scattering (DLS)

Both size (mean hydrodynamic diameter, Z-Average) and size distribution (polydispersity index, PDI) of the liposomal formulations were assessed by means of dynamic light scattering (DLS). DLS measurements were performed using a Zetasizer Nano ZS, Malvern Instruments Ltd. (Malvern, Worcestershire, UK), in back-scattering mode at an angle of 173°. Beforehand, liposomes were diluted to 0.25 mg PL per mL with sterile saline and equilibrated at 25 °C. Size measurements were performed in quintuplicate. Data analysis was performed with Zetasizer Software 6.30, Malvern Instruments Ltd. (Malvern, Worcestershire, UK).

#### 2.2.3. Cryo-Transmission Electron Microscopy (Cryo-TEM)

For cryo-TEM, vitrified specimens were prepared by a blotting procedure, performed in a chamber with controlled temperature and humidity using an EM GP-grid-plunger from Leica (Wetzlar, Germany) [40]. In total, 3 µL of the sample dispersion (c = 1 mg/mL) was placed onto an EM-grid coated with a holey carbon film (C-flat; Protochip Inc., Raleight, NC, USA) and frozen. Specimens were examined at 120 kV with a Libra 120 Plus TEM, Carl Zeiss Microscopy GmbH (Jena, Germany). The microscope was equipped with a Gatan 626 cryotransfer system and with a BM-2k-120 Dual-Speed on axis SSCCD-camera, TRS (Moorenweis, Germany).

#### 2.2.4. Zeta Potential Measurements

The zeta potential was determined by means of electrophoretic light scattering (ELS). ELS measurements were performed using a Zetasizer Nano ZS, Malvern Instruments Ltd. All samples were analyzed at 25 °C and measured in quintuplicate, being diluted either in glucose solution (5%; *w*/*v*; conductivity: 1.7–2.1 mS/cm), or in saline (conductivity: 16–20 mS/cm), both ensuring equal osmolality. Data analysis was performed with Zetasizer Software 6.30, Malvern Instruments Ltd. (Malvern, Worcestershire, UK).

#### 2.2.5. Mice

All in vivo protocols were approved by local authorities of Saxony-Anhalt, Germany, and complied with the guidelines of the Federation for Laboratory Animal Science Associations (FELASA) [43,44]. In order to avoid fluorescence signal absorption and scattering by hairs and bulbs of colored mice, hairless immunocompetent SKH1 mice (2 to 11 months old) with albino background were used, bred and kept in individually ventilated cages in groups of 2–5 individuals under specific pathogen-free conditions, at 12 h light/dark cycle. Standard diet and water were provided ad libitum. To ensure reproducibility and animal welfare, careful handling [45] and provision of basic cage enrichment (bedding, house, chewing sticks, nesting material) [46,47] were respected.

#### 2.2.6. Experimental Design for the Evaluation of In Vivo and Ex Vivo Biodistribution

Characterization of liposome biodistribution in mice was performed by means of in vivo and ex vivo fluorescence imaging (FI). Both female and male mice were investigated (n = 6/sex). The experimental protocol is presented in Figure 1. Each liposomal formulation was tested in n = 12 mice with 4 individuals per ex vivo endpoint (1 h, 24 h and 48 h). Initially, each mouse received 100 µL of liposomes (5 mg/mL PL in isotonic saline) i.v. into the tail vein. At defined time points, mice were narcotized and imaged (compare Section 2.2.6). After end point in vivo FI, while still being under narcosis, mice were euthanized by cervical dislocation. Organs were excised, washed in PBS and placed in two 12-well plates for ex vivo FI. The organ positions are shown in Appendix A.

#### 2.2.7. Fluorescence Imaging (FI)

For in vivo and ex vivo FI, the IVIS Spectrum FI system (PerkinElmer, Inc., Waltham, MA, USA) was used. Mice were anaesthetized by inhalation anesthesia (initially: 2.5% *v*/*v* isoflurane (Forene, Abbott, Wiesbaden, Germany) in oxygen at 3 L/min, maintenance: 2.5% at 0.3 L/min) in a XGI-8 narcosis system, Caliper Life Sciences (Runcorn, Cheshire, UK) and imaged at 37 °C in the IVIS Spectrum FI system. Images were taken in 2D mode. The FI system was equipped with a 150 W quartz wolfram halogen lamp. Gray scale and FI signals were recorded with a 4.1-megapixel (2048 × 2048) CCD camera at a working temperature of −90 °C. The respective experimental parameters are presented in Appendix A. Analysis of in vivo and ex vivo images was performed with Living Image^®^ software, PerkinElmer, Inc. (Waltham, MA, USA). For in vivo studies, the region of interest (ROI) was defined as an ellipsoid area (longitudinal axis 9.7 cm, transversal axis 4.47 cm) in the ventral perspective and for ex vivo studies as a 4 × 3 grid (each area 2.65 cm per axis). Total radiant efficiency (TRE) was assessed. It takes into account the exposure time and area, quantity of detected photons, a fixed spatial angle (steradian) and the exposure intensity, allowing quantitative comparisons between different mice and time points. The TRE of blank mice (as determined in untreated control mice) was subtracted from the TRE of treated mice. For in vivo normalization, TRE at t = 0.25 h was defined as 100% value for the individual time point and mouse. Ex vivo, the excised liver (where the highest TRE occurred) served as reference and for normalization of single organ TRE.

#### 2.2.8. Statistical Analysis

The calculation of absolute and relative values, arithmetic means and absolute and relative standard deviations was carried out using Microsoft Office Excel 2016, Microsoft Corporation (Redmond, WA, USA). Graphs were prepared using Origin 8.5.1, OriginLab Corporation (Northampton, MA, USA). All mice used in this study were included in subsequent analysis. No outlier values were discriminated. Since all groups were analyzed analogously, no blinding of analysts was conducted. Due to the robustness and longitudinal quality of the fluorescence readout, a sample size of up to 6 mice per gender and endpoint was estimated as sufficient.

## 3. Results

### 3.1. Physico-Chemical Characterization Showed Defined Size and Charge of DiR-Loaded Liposomes

Aiming to develop liposomes that circumvent the common problem of rapid systemic clearance of charged nanodispersions by the MPS, PS- and PG-enriched liposomal formulations were investigated. Before assessing their biodistribution kinetics in vivo, we thoroughly characterized the different formulations physico-chemically.

A detailed description of the developed systems has been published before [40]. Conventional liposomes contained either 30% (*w*/*w*) DOPS or 30% (*w*/*w*) DOPG, both in S100 (PC). “Stealth” liposomes comprised an additional 20% (*w*/*w*) PEG. S100 and S100 containing 20% (*w*/*w*) PEG served as reference. All liposomal formulations were investigated in terms of size, size distribution and surface charge, which was measured as zeta potential. Data are presented in Table 1.

The nanoformulations showed mean hydrodynamic diameter (Z-Average) of approximately 130 nm with narrow size distribution (PDI between 0.06 and 0.08) [48]. Representative DLS histograms are provided in Appendix A. Surface charge measurements were performed in two different iso-osmole dispersion media of high and low ionic strength to investigate the impact of ion strength on the zeta potential. Different zeta potential values were obtained, being less negative in saline than in glucose 5%. Furthermore, the individual liposomal formulations showed differences in their zeta potential values. The addition of the anionic PLs, PS or PG, led to a decrease in the zeta potential (becoming more negative), whereas the addition of PEGylated PLs increased the measured values towards more neutral values. For S100, the surface charge was close to zero, due to its zwitterionic character of PC. The addition of PEGylated PLs to S100 decreased the zeta potential slightly.

Taken together, liposomes were well characterized, defined in size and charge (neutral: S100, S100 PEG; anionic: PS, PG, PS PEG, PG PEG) and were consequently evaluated in vivo in a pre-clinical animal model.

### 3.2. In Vivo Liposome Biodistribution Differed in Biodistribution and Abdominal Accumulation, Depending on Both Formulation and PEGylation

Hence, we assessed the biodistribution by noninvasive optical imaging. Kinetics and spatial patterns of both conventional (S100, PS, PG) and “stealth” liposomal formulations (S100 PEG, PS PEG, PG PEG) in healthy mice were investigated in and ex vivo. The experimental protocol is presented in Figure 1. Representative full body images of female mice in the ventral perspective are presented in Figure 2, allowing qualitative comparisons.

DiR fluorescence values tended to be higher in female (Figure 2) than in male mice (Appendix A), probably due to inter-group age and weight differences, combined with the fixed dose of 0.5 mg PL per mouse. Therefore, we analyzed and presented the data separated by sex, even though qualitative observations were congruent.

S100 liposomes showed only slight accumulation in the cranial abdomen, while S100 PEG liposomes distributed homogeneously and at stable signal intensities over time. Contrastingly, for PS liposomes a signal maximum was determined at 0.25 h, predominantly in the cranial abdomen. PS PEG liposomes on the other hand dispersed more disseminated at t = 0.25 h and 1 h, showing a bifocal abdominal accumulation, probably representing liver and spleen. Both PS formulations were rapidly cleared.

PG liposomes were observed only at weak fluorescence intensities already from t = 0.25 h onwards; only slight accumulation in the cranial abdomen was seen. At t = 24 h and 48 h, the PG liposome signal appeared to be more homogenously distributed than before, indicating a biphasic mobilization process of liposomes. As compared to PG liposomes, PG PEG liposomes showed a higher abdominal signal intensity and more disseminated signal distribution at t = 0.25 h, combined with a noticeable signal decrease over time.

Taken together, the tested liposomal formulations showed sharply contrasting distributions, with S100 PEG liposomes circulating evenly throughout the mouse body at stable signal intensities over time, while S100, PS, PS PEG and PG PEG liposomes accumulated abdominally and dissipated rapidly. PG liposomes appeared steadily at low intensities. At t = 0.25 h, PEGylated formulations tended to be more systemically disseminated than conventional liposomes.

### 3.3. Quantification of In Vivo Fluorescence Signal Showed Prolonged Circulation of S100PEG and Fast Clearance of Other Formulations

For quantitative comparison of the differences in liposome distribution and pharmacokinetics demonstrated in Figure 2, the TRE of each mouse was normalized to its respective value at t = 0.25 h. The relative TREs for female mice are presented in Figure 3 and for male mice in Appendix A.

S100 liposomes showed relative TRE over 50% up to 3 h post injection, which further decreased over time. PS, PS PEG and PG PEG formulations presented a faster reduction in relative TREs, already being reduced to relative TRE < 50% at t = 1 h, especially in female mice. In male mice, the reduction of both S100 and PS liposomes was less pronounced, conceivably due to signal loss due to higher body weight of male mice. In contrast, relative TREs of PG liposomes decreased between t = 1 h and t = 24 h, possibly already having reached the post-logarithmic phase of elimination kinetics at t = 0.25 h and was followed by slow decrease. This profile was observed consistently in both sexes, suggesting a biphasic mobilization process. For PG PEG liposomes, relative TRE values decreased more than for PG liposomes, which may be due to the normalization of t = 0.25 h and the already substantially reduced absolute value of PG liposome TRE at that time point. Strikingly, S100 PEG preserved a relative TRE of approximately 80% for up to 48 h.

Thus, S100 PEG liposomes show stable circulation duration and low clearance rate, confirming the qualitative assessment of Figure 2. All other formulations are substantially eliminated by t = 24 h. It can be concluded that the intended prolonged circulation time due to the introduction of MPEG-2000-DSPE as a stealthy, hydrophilic and flexible shell was very successful to prolong the circulation time for S100 liposomes but showed only slight influence on the biodistribution of PS and PG liposomes. Having obtained a first overview of biodistribution, we moved on to more in-depth organ-based FI analysis.

### 3.4. Quantification of Ex Vivo Single Organ TREs Shows Systemic Accumulation of S100 PEG and Fast Clearance of Other Formulations via Liver and Spleen

For specification of the longitudinal systemic in vivo biodistribution of liposomal formulations in healthy mice, we continued with ex vivo examination of single organs. Representative qualitative images are presented in Appendix A. For quantitative purposes, TREs of single organs were normalized to the respective liver value of each mouse, since the highest TREs were detected in livers for all liposomal formulations.

Between female (Figure 4) and male (Appendix A) mice, no consistent differences of the individual organ liposome pharmacokinetic profiles were observed. All relevant organs were probed, in most of which no DiR signal was detectable. Therefore, we presented only a selection in Figure 4 and Appendix A; an overview about all organs can be found in Appendix A, for female and male mice, respectively. Over all endpoints, conventional liposomes showed accumulation predominantly in liver and to some extent in spleen (Figure 4), while PEGylated formulations accumulated in the liver and moderately in spleen. In some cases, slight fluorescence signals were also recorded in the uterus and ovaries. The accumulation of nanocarriers in the ovaries has been observed before for polymer [1,49] and lipid nanocarriers [50]. In blood samples, practically no DiR signal was detected at t = 1 h, except for S100 PEG liposomes, where some fluorescense intensities were found. This indicates a clearance from the blood circulation within the first hour post injection.

Impressively, S100 PEG liposomes were systemically distributed, at moderate to high relative TREs in several organs (blood, heart, lungs, spleen, kidneys and uteri/ovaries) at all endpoints.

In summary, conventional liposomes were rapidly eliminated from the whole circulation of the mouse body, showing only residual fluorescence intensities in liver, spleen and to some extent in the ovaries/uterus. PEGylation led to a differential elimination dynamic involving both liver and spleen, whereas differences between PL endure. S100 PEG had a long circulation duration and distribution through most organs over at least 48 h post injection.

## 4. Discussion

For both anionic PLs, PS [13] and PG [28,29], immunomodulatory properties have been demonstrated in models of wound healing and inflammation. Former studies by our group elucidated their ability to decrease the production of TNFα in a mIFNγ/LPS stimulated mouse peritoneal macrophages assay in vitro [40]. In the context of potential clinical application, e.g., in the treatment of inflammation after myocardial infarction [20], we aimed to characterize the biodistribution of PS- and PG-enriched nanodispersions in a pre-clinical mouse model and evaluate the results in context with PC nanoformulations. With emphasis on the comparison of the PLs, DiR-loaded PL-enriched nanodispersions were prepared and administered systemically in healthy mice. Liposome biodistribution was assessed by means of in and ex vivo FI. As previously confirmed in vitro, neither cytotoxic nor hemolytic properties were detected, and i.v. injected liposomes did not cause observable side effects in vivo.

In nanomedicines, size, size distribution and surface characteristics (charge, chemistry) are crucial physicochemical parameters, determining their pharmacokinetics and clearance rates in vivo, thereby being our primary analysis of PS- and PG-enriched nanodispersions. DiR-loaded PS- and PG-enriched liposomes were prepared by extrusion technique and revealed mean hydrodynamic diameter smaller than 150 nm with PDI values of less than 0.1, both indicating defined properties and highly monodisperse size-distributions. The addition of the negatively charged PLs, PS and PG, did not alter morphological properties of the liposomes, as was described previously [40]. To support this, in Appendix A, cryo-TEM images of all investigated liposome formulations are presented. Here, only PEG-related improvements of vesicle structure due to an increase in saturated fatty acids were expected and seen [40]. Furthermore, PS and PG addition led to a decrease in their zeta potential values. For both, comparable zeta potentials were observed. In the case of conventional PS and PG liposomes, the zeta potential was more negative than for “stealth” liposomes. The addition of PEGylated PLs increased the zeta potential values of PS PEG and PG PEG liposomes. For the neutrally charged S100, the decrease in zeta potential is due to minor amounts of anionic phospholipids in the PEG-shield providing phospholipid MPEG-2000-DSPE [6]. In addition, the presence of MPEG-2000-DSPE could also change the charge distribution (and compensation) and the shear plane, which would also result in a different zeta potential. Although all nanodispersions were loaded with the fluorescent dye DiR, their physicochemical properties were comparable to previously reported results [40]. Thus, these liposomes were suitable for biodistribution studies in a pre-clinical mouse model. For i.v. injection, liposomes were freshly prepared to reduce potential stability impacts.

The impact of particle size, charge, surface curvature and other factors on the biodistribution of nanomaterials has been reviewed by Moghimi et al. [37]. In our study, conventional and “stealth” liposomes presented comparable sizes but revealed different biodistribution profiles. Conventional liposomal formulations presented fast elimination kinetics (PS and PG clearance: at least 50% within 1 h), which is in agreement to literature values [51,52]. Literature suggests that the circulation time also depends heavily on the phase transition temperature of the used PLs [53,54]. As we used the natural, unsaturated S100, a PL with low-phase transition temperature [40], we expected a somewhat longer circulation, e.g., 80% recovery 2 h, as it was described by Semple et al. [53]. Especially in the investigated female mice, S100 liposomes resulted in 75% recovery after 1 h, indicating a relatively long circulation for the usually rapidly cleared conventional liposomes [51].

As expected, the addition of PEG, especially to S100, led to a prolongation of liposome circulation. “Stealth” S100 liposomes were distributed throughout all organs, whereas conventional S100 liposomes were enriched in liver and spleen. Hydrophilic surface modification by the addition of PEG resulted in prolonged circulation, which was comparable to marketed products like Doxil™/Caelyx™ [6,51]. Contrastingly, PS and PG conventional and “stealth” liposomes presented fast elimination kinetics, while accumulating in organs rich in cells of the MPS. Most likely, this is due to PS’ and PG’s recognition and immediate phagocytosis by cells of the MPS, as these cells are predominantly localized in both organs. Comparative in vivo studies examining the biodistribution of PS and PG are rare. In 1992, Gabizon and Papahadjopoulos systematically investigated PL surface charges and their impact on liposome clearance [55]. There, either unsaturated PS or unsaturated PG, both with comparable molar ratios, was formulated in unsaturated PC and cholesterol, and then sized and injected i.v. into female mice. Mice were euthanized after 4 h and organ radioactivity was quantified by a γ-counter. The authors showed a rapid clearance of PS- and PG-enriched liposomes from the blood stream and accumulation in liver and to some extent in the spleen. Slight differences in liposome organ distribution were observed between PS and PG. Predominantly, PS was localized in the liver (75% RD/organ) and in the spleen (10% RD/organ), whereas PG also showed radioactivity in the blood (9% RD/organ) and lower radioactivities in the liver (43% RD/organ) and spleen (1% RD/organ).

Our ex vivo images (Appendix A) also presented differences between the TREs of PS- and PG-treated organs. Here, it seemed that PS more potently accumulated in liver and spleen than PG. This can be explained by different phagocytosis rates and mechanisms, as PS can be engulfed by cells of the MPS and others, e.g., by hepatocytes [56] or fibroblasts, whereas PG is only phagocytized by monocytes and macrophages [56]. Furthermore, we previously observed PS being more potently engulfed than PG in vitro [42]. The variation in their phagocytosis activities may also explain the differences between the in vivo biodistribution profiles of PS and PG. For PG, it seemed that a time-dependent redistribution of the liposomes occurred, probably due to slower accumulation in the liver compared to PS.

The addition of PEG to anionic liposomal formulations did not result in prolonged liposome circulation in vivo, while ex vivo higher accumulation in the spleen was observed. PEGylation of negatively charged liposomal formulations is not common. One study by Levchenko et al. systematically investigated PS-enriched liposomes with different types of PEG, either PEG 750 or PEG 5000 [57]. There, they showed that addition of PEG 750 was not suitable to overcome the PS phagocytosis by cells of the MPS, whereas PEG 5000 containing PS liposomes revealed circulation duration comparable to liposomes lacking PS, resulting in a slight prolongation. This study is supported by Chiu et al., who, in addition to the characterization of the PEG chain length, also investigated the amount of PEG needed to overcome PS’ recognition by the MPS [58]. They found that a content of 15% of PEG 2000 successfully reduced plasma protein interactions in liposomes containing 10% PS. In our case, we used PEG 2000, only marginal impacts on PS’ or PG’s biodistribution in vivo were observed. As we used 30% of anionic PLs, probably higher amounts of PEG are necessary, but this may raise further concern, e.g., liposome to micelle transition by PEG, as was described by Chiu [58]. Thus, we only used 5% of PEG 2000. In addition, our results are in accordance with results observed by Boerman et al. [52], who described a rapid clearance of PS containing liposomes, which could not be overcome by the addition of PEG.

However, in our case, slightly higher TREs in spleens were detected ex vivo. Probably, the shield effect of PEG 2000, with medium chain length, is situated between PEG 750 and PEG 5000 [57], and therefore results in a slightly higher and prolonged accumulation in the spleen, as compared to conventional liposomes. Furthermore, recent studies on PEGylated lipid nanoparticles imply a PEG-shedding effect with subsequent increase in phagocytosis rate of Kupffer-cells [59,60], or via immune complex formation, which is complement and/or IgM mediated. Long acyl-chains in the PEG used in this study yield a higher potential to associate with complement or be the object of IgM opsonization, possibly explaining the increased recovery of PS PEG and PG PEG liposomes in organs rich of the MPS, namely, liver and spleen. Especially the increased splenic accumulation in PEGylated liposomes might be explained by this dynamic, including increased B cell mediated trapping.

## 5. Conclusions

In conclusion, we non-invasively investigated the impact of composition and PEGylation on the biodistribution of well-defined PC, PS and PG liposomes. PC liposomes, conventionally formulated, were rapidly cleared from the circulation, while PEGylation resulted in prolongation of liposome circulation. In contrast, PS and PG liposomes, both as conventional or PEGylated formulations, were rapidly cleared. Non-PEGylated PS and PG liposomes did accumulate almost exclusively in the liver. In contrast, PEGylated PS and PG liposomes were observed mainly in liver and spleen. In summary, PEGylation of PS and PG liposomes was not effective to prolong the circulation time but caused a higher uptake in the spleen. This observation raises new questions, which will be investigated in future studies. For example: “What is the threshold concentration for PS or PG to change the circulation time?” The present study was conducted with a PS:PC ratio of 3:7 and a PG:PC ratio of 3:7. If the uptake is receptor mediated, lower concentrations might be already effective in the change of the biodistribution. How general is the observed effect? This question will require studies in different mice strain or other species (e.g., rats) and investigations with other long circulation drug delivery systems (e.g., PS- or PG-doped PEG-PLGA nanoparticles). Future studies will be devoted to investigating PS- and PG-induced effects both on (1) biodistribution and (2) on their anti-inflammatory activity. The concentration dependency could be quite different for both effects.

## Figures and Tables

**Figure 1 pharmaceutics-13-00282-f001:**
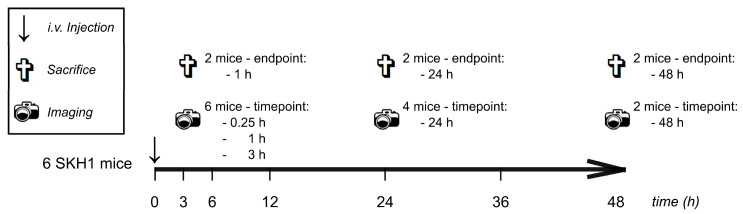
Experimental protocol for in vivo and ex vivo biodistribution study. Both female and male mice were investigated. For each sex, 6 mice were used, respectively.

**Figure 2 pharmaceutics-13-00282-f002:**
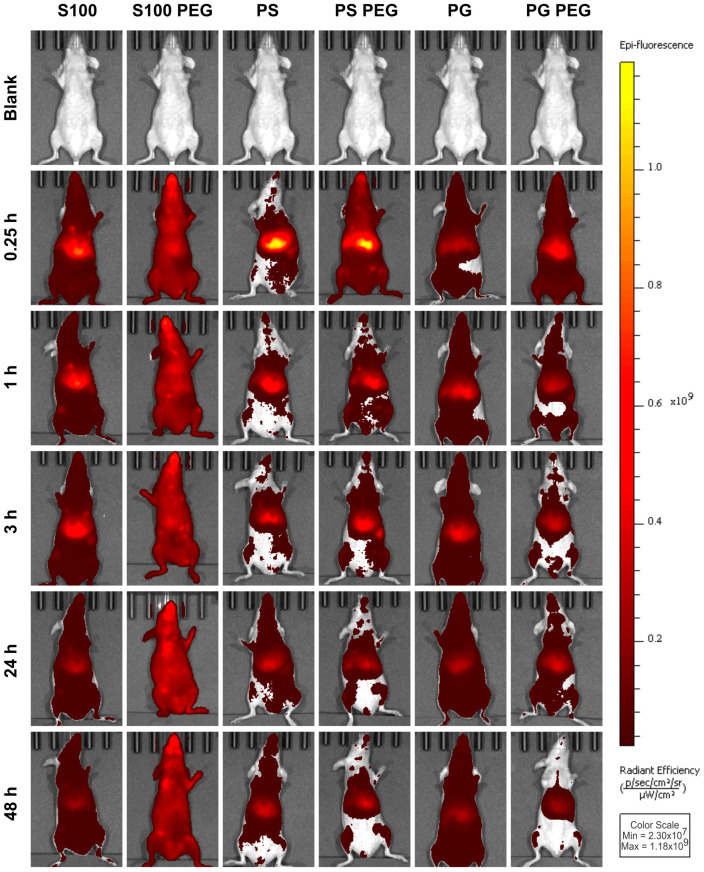
In vivo liposome pharmacokinetics differed in distribution and abdominal accumulation, depending on both formulation and PEGylation. DiR-loaded liposome-treated female mice (ventral perspective), as assessed by in vivo fluorescence imaging (FI). The total radiant efficiency (TRE) of representative mice of each group and time points is shown. For direct comparison, radiant efficiency was normalized.

**Figure 3 pharmaceutics-13-00282-f003:**
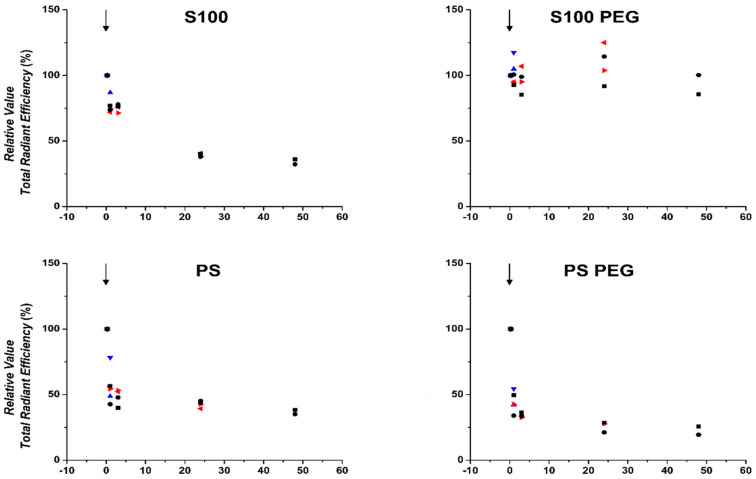
Quantification of in vivo fluorescence signal showed prolonged circulation of S100 PEG and fast clearance of other formulations. Signals were evaluated from defined region of interest (ROI) in DiR-loaded liposome-treated female mice (ventral perspective), as assessed by in vivo FI at 0.25, 1, 3, 24 and 48 h post injection. M1 to M6 represent individual mouse values. M1 and M2 were sacrificed at t = 1 h, M3 and M4 at t = 24 h and M5 and M6 at t = 48 h. Presented values are relative to t = 0.25 h.

**Figure 4 pharmaceutics-13-00282-f004:**
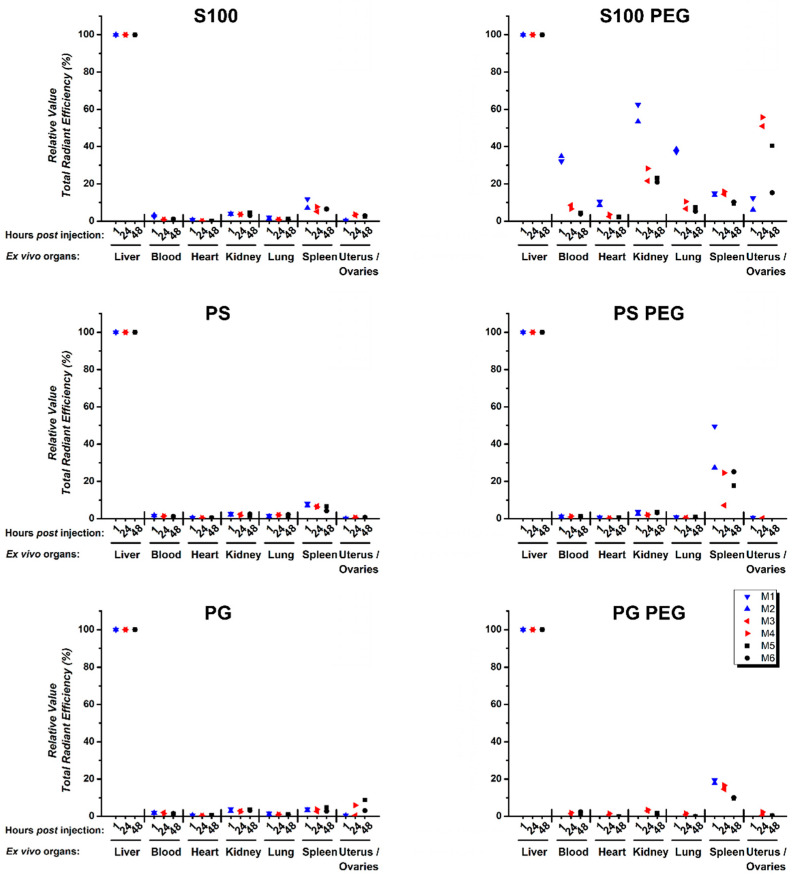
Quantification of ex vivo single organ total radiant efficiencies (TREs) (pre-selected) shows systemic and spatial accumulation of S100 PEG and fast clearance of other formulations via liver and spleen. Signals were evaluated from defined ROI in DiR-loaded liposome-treated excised single organs, as assessed by ex vivo FI. M1 to M6 represent individual female mouse values. M1 and M2 were sacrificed at t = 1 h, M3 and M4 at t = 24 h and M5 and M6 at t = 48 h. Presented values are relative to the TRE of the respective liver.

**Table 1 pharmaceutics-13-00282-t001:** Particle size (Z-Average), size distribution (polydispersity index, PDI) and zeta potential values of different liposomal formulations. Data are shown as mean and SD (n = 3).

Liposomal Formulation	Z-Average (nm)	PDI	Zeta Potential (mV) in Saline	Zeta Potential (mV) in Glucose 5%
S100	125 ± 6	0.06 ± 0.01	−0.9 ± 1.4	0.3 ± 0.3
S100 PEG	116 ± 13	0.06 ± 0.01	−2.5 ± 1.5	−9.1 ± 0.8
PS	125 ± 11	0.07 ± 0.01	−26.9 ± 1.3	−60.0 ± 3.9
PS PEG	115 ± 10	0.06 ± 0.01	−4.2 ± 2.0	−16.0 ± 1.4
PG	120 ± 4	0.08 ± 0.02	−27.1 ± 2.1	−57.8 ± 1.5
PG PEG	116 ± 13	0.06 ± 0.01	−3.7 ± 1.2	−16.1 ± 2.6

## Data Availability

Not applicable.

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
