# Peer review of "Towards the Development of Long Circulating Phosphatidylserine (PS)- and Phosphatidylglycerol (PG)-Enriched Anti-Inflammatory Liposomes: Is PEGylation Effective?"

_pharmaceutics, 2021, doi:10.3390/pharmaceutics13020282_

Round 1
Reviewer 1 Report
The authors were able to develop of long circulating liposomes with intrinsic antiinflammatory activity. PS- and PG-enriched liposomes were produced. Liposomes were formulated as conventional or PEGylated formulations. They had diameters below 150 nm, narrow size distributions and composition-dependent surface charges.
The authors wanted to get answers to the following questions:
- Are the biodistribution profiles of PS- and PG-enriched nanodispersions in healthy mice comparable?
- Does PEGylation of PS- and PG-enriched nanodispersions effect their biodistribution in healthy mice?
Through their experiments, the authors received answers to their questions, ie. they managed to clarify the formulated scientific problems.
Advantages:
Contribution to the clinical application of liposomal systems - very current research.
Application of non-invasive technique for biodistribution.
The experimental results fully support the discussion and conclusion.
Figure 2 is meaningful, and attracts attention.
A lack:
The term PEGylation is not explained in the introduction, it needs to be explained.
PEGylation liposome scheme, maybe it should show (especially the surface of the liposome).
Reviewer 2 Report
The manuscript “Towards the Development of Long Circulating PS and PG En- 2 riched Anti-inflammatory Liposomes: is PEGylation Effective?” by Klein et al is very well written with established rational and robust experimental design.
Rewrite the title to remove acronyms
More attention is need to the references/citations.
Why are the authors more interested in the biodistributions than the inflammatory effect it-self for PS an PG?
It would have been if SEM/TEM images of different liposomes can be shown.
How would these liposomes be taken, intravenously? Any comments on toxicity/biocompatibility?
Any difference in stability between three types of liposomes.
Reviewer 3 Report
The present manuscript deals with the investigation of the effect of pegylation on the biodistribution, accumulation and circulation time of liposomes. The interesting point is the direct comparision between phospholipids prepared with only phosphatidylcholine or the mixture phosphotidylcholine/phosphatidlserine and phosphotidylcholine/phosphatidylglycerol that are pegylated or not. The authors conclude that pegylation prolongs the circulation time effectively only for liposomes prepared with phosphatidylcholine but it caused accumulation in the spleen for the other liposomes prepared with phosphatidylcholine and phosphatidylserine. This conclusion is supported by the presented results. However, it should be carefully examined if it is a general conclusion or it is related only at the experimental conditions tested both as regard liposomes and the in vivo model.
Please amend the text from the sentence "Error! Reference source not found". Sometimes, the presence of this message strongly affects the readability of the manuscript.
Despite the authors have evaluated the potential anti-inflammatory therapeutical effect of liposomal formulation containing PS and PG in a previous work, the term anti-inflammatory is not pertinent in the title, since the anti-inflammatory properties are not addressed in the present manuscript.
Page 10 line 360 It is not clear the sentence "possibly due to anionic PL residues in PEG".
Page 10 Line 367 The different effect of Pegylation in terms of the characteristics of the phospholipid should be better discussed to explain the relevance difference observed in circulation time and biodistribution.
Reviewer 4 Report
The manuscript pharmaceutics-1095130 aims to investigate the efficacy of PEGylation to prolong liposome circulation time.
The manuscript is well organized, the methodologies and the results are properly described. The discussion part in its overall is exhaustive.
However, before to be considered for publication minor revision is required;
1) Introduction can be reduced, avoiding long description and focusing more on the importance of the study. The study does not provide any new information, but supports the results already published in other studies in the field of PEGylation and circulation time.
2) In various part of the manuscript references are not available. I guess something wrong.
3)line 208-214 : This part is related to methodologies. Few words to introduce the meaning of performing the experiments should be reported before the illustration of the results
4) line 219 It is more appropriate to write an interval instead of PDI >0.1
5) Figure 4. The number reported on the x-axes should be bigger in font
Round 2
Reviewer 3 Report
The manuscript is suitable for pubblication